# Causality Meets the Table: Debiasing LLMs for Faithful TableQA via Front-Door Intervention

**Zhen Yang**[1,2,3], **Ziwei Du**[1,2,3], **Minghan Zhang**[1,2,3], **Wei Du**[1,2,3],
**Jie Chen**[1,2,3], **Fulan Qian**[1,2,3], **Shu Zhao**[1,2,3]*
[1]School of Computer Science and Technology, Anhui University
[2]Information Materials and Intelligent Sensing Laboratory of Anhui Province, Anhui University
[3]Anhui Provincial Key Laboratory of Security Artificial Intelligence, Anhui University
*zhaoshuzs2002@hotmail.com

## Abstract

Table Question Answering (TableQA) combines natural language understanding and structured data reasoning, posing challenges in semantic interpretation and logical inference. Recent advances in Large Language Models (LLMs) have improved TableQA performance through Direct Prompting and Agent paradigms. However, these models often rely on spurious correlations, as they tend to overfit to token co-occurrence patterns in pretraining corpora, rather than perform genuine reasoning. To address this issue, we propose **C**ausal **I**ntervention **T**ableQA (CIT), which is based on a structural causal graph and applies front-door adjustment to eliminate bias caused by token co-occurrence. CIT formalizes TableQA as a causal graph and identifies token co-occurrence patterns as confounders. By applying front-door adjustment, CIT guides question variant generation and reasoning to reduce confounding effects. Experiments on multiple benchmarks show that CIT achieves state-of-the-art performance, demonstrating its effectiveness in mitigating bias. Consistent gains across various LLMs further confirm its generalizability. We release our code here.

## 1 Introduction

Tabular data is a prevalent type of structured information, commonly found in many fields [Yang et al., 2025, Lee et al., 2024, Xia et al., 2023]. Table Question Answering (TableQA), which aims to answer natural language questions over tables, plays a key role in decision support and data analysis. Early methods focused on SQL-based semantic parsing [Zhong et al., 2017] or pretraining on table-specific corpora [Ou and Liu, 2022, Eisenschlos et al., 2020, Xie et al., 2022]. More recently, Large Language Models (LLMs) have achieved strong results on TableQA by leveraging In-Context Learning [Sui et al., 2023, Chen, 2023a] and Chain-of-Thought (CoT) prompting [Cheng et al., 2023, Ye et al., 2023b]. Based on CoT, two main paradigms have emerged: Direct Prompting (DP), which performs natural language reasoning, and Agent, which relies on symbolic code execution.

Despite strong empirical performance, prompting strategies in LLM-based TableQA are often not robust [Ye et al., 2023a]. LLMs tend to rely on token co-occurrence patterns from pretraining data, which can lead to spurious correlations and unfaithful reasoning [Lyu et al., 2023, Wang et al., 2023c, Bao et al., 2024, Turpin et al., 2023]. For example, as shown in Figure 1, phrases like exactly frequently co-occur with answers like yes, causing LLMs to prefer yes even when no is correct.

---

*Corresponding authors

39th Conference on Neural Information Processing Systems (NeurIPS 2025).

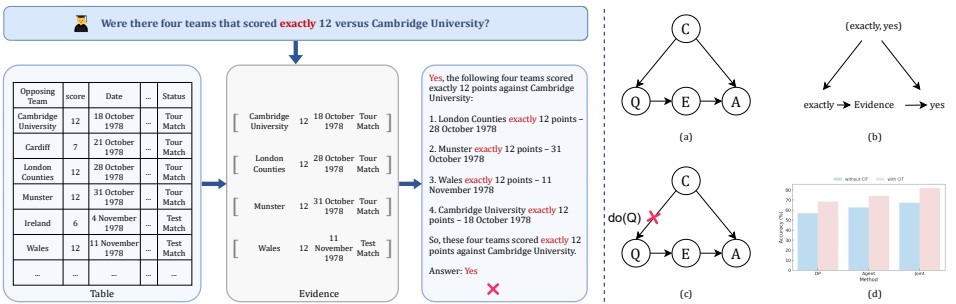

Figure 1: Illustration of token co-occurrence bias as a confounder in LLM-based TableQA. (a) Causal graph showing confounding in reasoning; (b) Real-world example where pretraining bias causes unfaithful answers; (c) Do-intervention to block the back-door path; (d) Adversarial example demonstrating spurious correlations.

This creates a confounder that affects both question interpretation and answer prediction, forming a back-door path that distorts reasoning. Although causal methods have been proposed to address such issues [Niu et al., 2021, Tian et al., 2022, Guo et al., 2023], they often rely on observable confounders or model internals, limiting their applicability in TableQA where confounders are typically latent.

Beyond the above qualitative analysis, we also conduct quantitative verification of these limitations. Specifically, we apply double negation perturbations to logically equivalent examples from TabFact dataset [Chen et al., 2020] and observe a substantial drop in accuracy. This suggests that LLMs rely more on surface-level linguistic patterns than on deep reasoning.

To address these challenges, we reinterpret TableQA from a causal perspective. As shown in Figure 1(a), ideal reasoning follows $Q \rightarrow E \rightarrow A$, but token co-occurrence introduces a confounder $C$, forming a spurious path $Q \leftarrow C \rightarrow A$. We adopt front-door adjustment [Pearl et al., 2016], which enables causal estimation using only observed variables, without requiring access to $C$ or model internals. We propose Causal Intervention TableQA (CIT), a framework that mitigates bias from token co-occurrence patterns through front-door adjustment. CIT avoids explicit do-calculus by decomposing the adjustment process into four components: (1) *Question Variant Generation*, which produces diverse paraphrases of $Q$ to reduce lexical bias; (2) *Evidence Aggregation*, which combines retrieved content across variants to improve coverage; (3) *Answer Inference*, which applies both DP and Agent reasoning strategies; (4) *Joint Voting*, which selects the final answer via majority voting. Experiments across multiple TableQA benchmarks and LLMs show that CIT consistently improves reasoning robustness and generalization. Our main contributions are as follows:

- **Causal formulation of co-occurrence bias**: We are the first to introduce causal intervention into LLM-based TableQA by modeling token co-occurrence as latent confounding and applying front-door adjustment to mitigate its effect.

- **Efficient intervention via question variants:** We estimate causal effects using semantically diverse question variants. A single-pass generation strategy ensures low overhead.

- **Broad empirical validation:** CIT achieves state-of-the-art performance on multiple datasets across both open- and closed-source LLMs, demonstrating strong generalization.

## 2 Related Work

### 2.1 LLM-based TableQA

Recent advances in large language models (LLMs) have greatly improved TableQA performance by leveraging general reasoning capabilities [Pal et al., 2023, Lee et al., 2024, Zhong et al., 2017, Yang et al., 2025]. Existing methods mainly follow two paradigms: Direct Prompting, which guides reasoning in a single step [Sui et al., 2023, Chen, 2023a], and Agent, which decomposes the task into symbolic operations [Li et al., 2024b, Lei et al., 2023]. Representative methods are listed in Appendix A. However, these approaches focus primarily on guiding LLM reasoning, while

overlooking a key issue: LLMs often encode token co-occurrence patterns from pretraining data, which can induce spurious correlations between the question and the answer.

## 2.2 Causal Intervention

Causal inference offers a principled framework for addressing bias through interventions [Pearl et al., 2016, Pearl, 2019, Ren et al., 2023a,b]. Prior work has applied counterfactual [Niu et al., 2021, Xu et al., 2023, Yang et al., 2023b], back-door adjustment [Tian et al., 2022, Zhu et al., 2023], and front-door adjustment [Yang et al., 2021, Zhang et al., 2024a, Yang et al., 2023a] to mitigate spurious correlations. Recent studies have extended ideas to LLMs [Jin et al., 2023, Lyu et al., 2024], although many rely on heuristics or simplified causal graphs [Wang et al., 2023b, Tang et al., 2023]. In contrast to back-door methods that require explicit modeling of confounders, which is often infeasible for LLMs, front-door adjustment enables causal estimation using only observed variables. This makes it particularly suitable for LLM-based TableQA.

# 3 Preliminaries

## 3.1 TableQA

Given a question $Q$ and a table $T$, an LLM first interprets the question, retrieves relevant evidence $E \subseteq T$ based on $Q$, and then reasons over $Q$ and $E$ to produce the final answer $A$. This process can be formally expressed as Equation 1:

$$E = Prompt_{retrieve}(Q, T), \quad A = Prompt_{answer}(Q, E) \tag{1}$$

Here, $Prompt_{retrieve}$ refers to the step where the LLM selects evidence based on the question. $Prompt_{answer}$ performs reasoning over the question and the evidence to get the answer.

## 3.2 Structural Causal Model (SCM)

Causal inference offers a framework for modeling interventions and estimating causal effects. A central tool is the Structural Causal Model (SCM)[Pearl et al., 2016], which represents dependencies among variables as a directed acyclic graph (DAG) $G = (V, E)$, where $V$ is the set of nodes and $E$ is the set of edges. In Figure 1, we illustrate the causal graph constructed for the TableQA task and explain its components as follows.

$Q \to E \to A$. In TableQA, the input question $Q$ determines the selection of evidence $E$, which in turn leads to the answer $A$. This forms the ideal causal path $Q \to E \to A$.

$Q \leftarrow C \to A$. During pretraining, LLMs tend to overfit to token co-occurrence patterns in the corpus. In certain cases, this behavior interferes with reasoning and leads to biased predictions. We treat this as a latent confounder, denoted as $C$, which introduces a spurious back-door path $Q \leftarrow C \to A$.

$do(Q)$. To identify the true causal effect, it is necessary to block the influence of the confounder $C$ using the do-operator [Fenton et al., 2020]. In an SCM, applying $do(Q)$ corresponds to removing all incoming edges to $Q$, thereby eliminating the indirect effect of $C$ on $A$ through $Q$. In causal inference, the do-operator represents an ideal intervention that forcibly sets the value of a variable. However, such interventions are typically infeasible in observational data. Therefore, techniques such as front-door or back-door adjustment are commonly used to estimate causal effects without directly applying the do-operator.

## 3.3 Front-door Adjustment

Back-door adjustment requires access to the confounder $C$, which is unobservable in LLMs. In contrast, front-door adjustment bypasses this need and is thus more applicable. By the law of total probability, the interventional distribution is given in Equation 2.

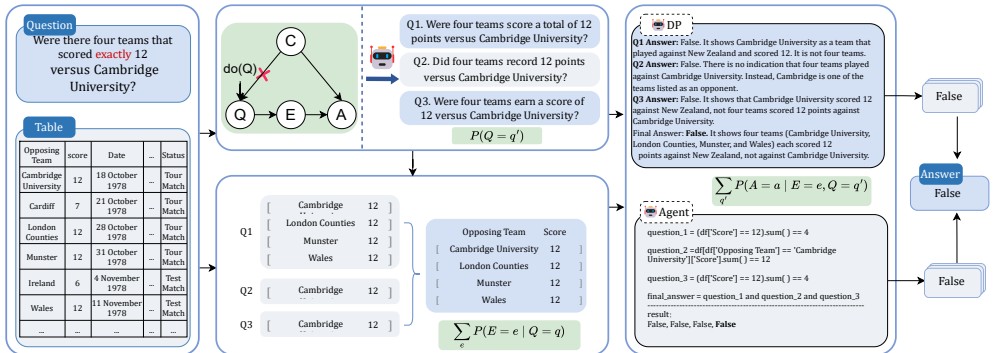

Figure 2: Overview of the CIT. Given a question and table, CIT derives the final answer through: (1) Question Variant Generation, which produces semantically diverse variants; (2) Evidence Aggregation, which extracts and unifies evidence across variants; (3) Answer Inference, which combines Direct Prompting and Agent; and (4) Joint Voting, which selects the final answer via majority voting.

$$P(A \mid do(Q)) = \sum_e P(A = a \mid do(Q = q), E = e)\, P(E = e \mid do(Q = q)) \tag{2}$$

By applying the law of total probability and the assumptions of SCM, this expression can be further transformed into the final formula shown in Equation 3. The full derivation is provided in Appendix B.

$$P(A = a \mid do(Q = q)) = \sum_e P(E = e \mid Q = q) \sum_{q'} P(A = a \mid E = e, Q = q')\, P(Q = q') \tag{3}$$

## 4 Method

By applying the do-operator, we block the bias introduced by the confounder $C$. Through front-door adjustment, this intervention can be reformulated into a tractable expression using only observed variables. Based on Equation 3, our method consists of four components: (1) Question Variant Generation, (2) Evidence Aggregation, (3) Answer Inference, and (4) Joint Voting. We describe each component in detail below.

### 4.1 $P(Q = q')$: Question Variant Generation

To estimate $P(Q = q')$, we generate a set of semantically equivalent question variants $\{q'_i\}_{i=1}^n$ that preserve the intent of the original question $Q$ while differing in surface form. To reduce the cost of LLM inference, we adopt a single-pass generation strategy using a prompt-based generator, denoted as $Prompt_{gen}$, which produces all variants in a single call, as shown in Equation 4. Here, $Q$ is the input question, $T$ is the table, and the prompt used for variant generation is detailed in Appendix C.

$$\{q'_i\}_{i=1}^n = LLM(Q, T, Prompt_{gen}) \tag{4}$$

Since all variants are generated simultaneously, their generation probability is approximated as uniform. As a result, $P(Q = q')$ is treated as a constant and omitted from the final formulation.

Beyond the causal perspective, our method can also be understood semantically. When encoded by an LLM, the original question and its variants are represented as high-dimensional vectors in a shared semantic space. Generating multiple variants effectively samples points around the original question, forming a dense semantic neighborhood. This encourages the LLM to reason over meaning rather than surface form, which improves robustness and helps mitigate bias.

## 4.2 $\sum_e P(E = e \mid Q = q)$: Evidence Aggregation

To estimate $\sum_e P(E = e \mid Q = q)$, we extract supporting evidence for the given question. Since CIT considers multiple question variants $\{q'_i\}$, evidence must be extracted for each variant. To reduce the cost, we adopt a separate single-pass strategy using $Prompt_{retrieve}$ for evidence extraction. All question variants are provided to the LLM in one prompt, and the LLM extracts the corresponding evidence $e_{q'_i}$ for each variant and then aggregate to form the final evidence, as shown in Equation 5. The prompt used by $Prompt_{retrieve}$ is detailed in Appendix C.

$$LLM(T, q, q', Prompt_{retrieve}) \rightarrow e_{q'_i}, \quad e = \bigcup_{i=1}^{n} e_{q'_i} \tag{5}$$

This union operation ensures that all potentially useful evidence across diverse paraphrases of the question is captured. By aggregating information from multiple linguistic perspectives, we construct a more complete and robust foundation for downstream reasoning.

## 4.3 $\sum_{q'} P(A = a \mid E = e, Q = q')P(Q = q')$: Answer Inference

To estimate $\sum_{q'} P(A = a \mid E = e, Q = q')P(Q = q')$, we infer under each variant $q'$. We consider two reasoning paradigms commonly used in LLM-based TableQA: Direct Prompting and Agent.

**Direct Prompting (DP) Reasoning.** DP guides the LLM to generate step-by-step reasoning via CoT to obtain the answer, improving performance on complex TableQA tasks. For each question variant $q'$, the LLM generates a token sequence conditioned on $q'$ and the retrieved evidence $e$, modeled autoregressively as Equation 6.

$$P(A = a^{DP} \mid E = e, Q = q') = \prod_{l=1}^{L} P(w_l \mid w_{<l}, e, q') \tag{6}$$

Here, $w_l$ is the $l$-th token, $w_{<l}$ the preceding tokens, and $L$ the sequence length. The LLM maximizes this joint probability to generate reasoning steps and the final answer $a^{\text{DP}}$. However, this formulation may overfit to frequent patterns from pretraining, leading to overconfident but unfaithful predictions. Our causal framework mitigates this by aggregating outputs across diverse question variants.

**Symbolic Reasoning with Agent.** In contrast to natural language reasoning, Agent allows the LLM to generate executable Python code for structured operations over table evidence. Given a question variant $q'$ and evidence $e$, the LLM produces a code snippet $code(q', e)$, and the final answer is obtained by executing it within a Python shell as Equation 7. Here $execute$ denotes run the code.

$$P(A = a^{Agent} \mid E = e, Q = q') = execute(code(q', e)) \tag{7}$$

This symbolic approach allows direct operations on tabular as filtering, aggregation, and arithmetic—enabling precise numerical. To reduce API cost, we adopt a unified execution strategy: the original question $q$ and its variants set$\{q'_i\}$ are processed in a single LLM call. The model reasons over each variant independently and aggregates intermediate results into a final answer $a^{\text{Agent}}$. This one-shot process improves efficiency and ensures semantic consistency across variants.

## 4.4 Joint Voting

CIT supports both DP and Agent reasoning, and integrates their outputs via majority voting to exploit their complementary strengths. For each question $q$, the framework performs $n$ rounds of DP and $m$ rounds of Agent reasoning, yielding answer sets $\mathcal{A}^{\text{DP}} = \{a_1^{\text{DP}}, \ldots, a_n^{\text{DP}}\}$ and $\mathcal{A}^{\text{Agent}} = \{a_1^{\text{Agent}}, \ldots, a_m^{\text{Agent}}\}$. The final prediction is selected by majority vote over both sets as Equation 8:

$$P(A = a \mid E = e, Q = q') = Majority\, Vote(A^{DP} \cup A^{Agent}) \tag{8}$$

In case of a tie, one of the top answers is selected uniformly at random. The hyperparameters $n$ and $m$ control the number of DP and Agent rounds. We vary $(n, m)$ in ablations to explore performance-cost

trade-offs. Beyond voting, this joint design enhances semantic diversity by reasoning over multiple variants $q'$ and evidence sets $E$. This is especially beneficial for ambiguous or biased questions, where aggregating diverse reasoning paths helps mitigate confounder-induced errors.

## 5 Experimental Setup

### 5.1 Datasets and Evaluation

**Dataset.** We evaluate on three datasets: WikiTableQuestions (WTQ) [Pasupat and Liang, 2015], TabFact[Chen et al., 2020], and FetaQA [Nan et al., 2022]. WTQ involves aggregation, comparison, and arithmetic reasoning, with 4,344 test examples. TabFact is a fact verification task over 2,024 samples. FetaQA features free-form questions that require integrating information on 2,003 samples.

**Evaluation.** Following prior work [Liu et al., 2024a, Yang et al., 2025], we use exact match accuracy for WTQ and TabFact, which focus on short-form answers. For FetaQA, which requires long-form generation, we report BLEU [Papineni et al., 2002] to evaluate answer quality.

### 5.2 Implementation Details

To ensure fair comparison, we first evaluate CIT using GPT-3.5 as LLM. To assess generalizability, we further test CIT across LLMs: *Open-source*: LLaMA 2-7B/13B/70B, DeepSeek-R1; *Closed-source*: GPT-3.5, GPT-4, GLM 4, Gemini 1.5, Claude 3.5. All LLMs temperature is 0.8.

Table 1: Results on WikiTableQuestions with GPT-3.5. CIT-DP and CIT-Agent show results for DP and Agent modes separately. CIT-DP&Agent shows the joint voting result.

| Method | Acc. |
| --- | --- |
| OmniTab (22' NAACL) | 61.30 |
| Codex SQL (23' ICLR) | 61.10 |
| BINDER (23' ICLR) | 64.60 |
| DATER (23' SIGIR) | 65.90 |
| DTE (23' ACL) | 54.20 |
| TACR (23' arXiv) | 60.20 |
| ITR (23' ACL) | 63.40 |
| StructGPT (23' EMNLP) | 57.00 |
| Liu et al. (24' arXiv) | 55.80 |
| Cabinet (24' ICLR) | 69.10 |
| CHAIN-OF-TABLE (24' ICLR) | 59.94 |
| ReAcTable (24' VLDB) | 68.00 |
| SYNTQA (24' EMNLP) | 70.40 |
| Mix-SC (DP&Agent) (24' NAACL) | 73.65 |
| TIDE (DP&Agent) (25' ICLR) | 75.00 |
| CIT-DP | 65.40 |
| CIT-Agent | 73.76 |
| **CIT-DP&Agent** | **76.38** |

Table 2: Results on TabFact dataset with GPT-3.5. CIT-DP and CIT-Agent show results for DP and Agent modes separately. CIT-DP&Agent shows the joint voting result.

| Method | Acc. |
| --- | --- |
| TAPAS-large (20' EMNLP) | 81.00 |
| TAPEX-large (21' ICLR) | 84.20 |
| SaMOE (22' ACL) | 86.70 |
| SASP (22' ACL) | 77.00 |
| T5-3B (22' EMNLP) | 83.68 |
| Codex end-to-end (23' ICLR) | 72.60 |
| Codex SQL (23' ICLR) | 80.70 |
| BINDER (23' ICLR) | 85.10 |
| DATER (23' SIGIR) | 85.60 |
| StructGPT (23' EMNLP) | 87.30 |
| CHAIN-OF-TABLE (24' ICLR) | 80.20 |
| ReAcTable (24' VLDB) | 86.10 |
| Tab-PoT (24' arXiv) | 85.77 |
| Mix-SC (DP&Agent) (24' NAACL) | 88.50 |
| TIDE (DP&Agent) (25' ICLR) | 89.82 |
| CIT-DP | 83.15 |
| CIT-Agent | 90.61 |
| **CIT-DP&Agent** | **91.30** |

### 5.3 Baselines

We compare CIT with pretraining models and LLM-based methods, include SASP [Ou and Liu, 2022], TAPAS-large [Eisenschlos et al., 2020], T5-3B [Xie et al., 2022], TAPEX-large [Liu et al., 2021], Task Configs [Chen et al., 2023], TARGET [Ji et al., 2024], TabCot [Chen, 2023b], TAG-QA [Zhao et al., 2023a], UniTabPT[Sarkar and Lausen, 2023], and Codex [Cheng et al., 2023], BINDER [Cheng et al., 2023], DATER [Ye et al., 2023b], StructGPT [Jiang et al., 2023], DTE [Wang et al., 2023a], TACR [Wu et al., 2023], ITR [Lin et al., 2023], Tab-PoT [Xiao et al., 2024], [Liu et al., 2024a],

CHAIN-OF-TABLE [Wang et al., 2024], ReAcTable [Zhang et al., 2024c], Cabinet [Patnaik et al., 2024], SYNTQA [Zhang et al., 2024b], [Liu et al., 2024b] and TIDE [Yang et al., 2025]. Details of the baseline implementations are provided in Appendix D.

# 6 Results and Analysis

## 6.1 Main Results

Table 1 shows that CIT achieves state-of-the-art performance on WikiTableQuestions, improving the previous best by 2.73%. On TabFact (Table 2), it outperforms the strongest baseline by 2.21%, a relative gain of 21.71% over the error rate. CIT also leads clearly on FetaQA (Table 3). We analyze the effectiveness of CIT from both qualitative and quantitative perspectives: *Qualitative Perspective:* CIT introduces causal reasoning via a structural causal model, enabling identification and blocking of confounding bias through front-door adjustment. Question variants help mitigate spurious correlations from pretraining, improving answer faithfulness. *Quantitative Perspective:* On adversarial data with double negation (Figure 1), CIT significantly outperforms non-intervention models, confirming its robustness. The combination of DP and Agent leverages complementary strengths and reduces reliance on any single reasoning mode.

Table 3: Results on FetaQA with GPT-3.5.

| Methods | BLEU(%) |
|---|---|
| Task Configs (23' ACL) | 27.80 |
| T5-large (23' SIGIR) | 30.54 |
| TAG-QA (23' ACL) | 31.84 |
| UniTabPT (23' NeurIPS) | 33.12 |
| Codex (23' SIGIR) | 27.96 |
| DATER (23' SIGIR) | 30.92 |
| TabCot (23' EACL) | 29.36 |
| TARGET (24' NeurIPS) | 24.13 |
| ReAcTable (24' VLDB) | 30.43 |
| CIT-DP | 36.15 |
| CIT-Agent | 33.65 |
| **CIT-DP&Agent** | **36.34** |

Table 4: Impact of answer selection in CIT-DP and CIT-Agent.

| Agent | DP | WTQ | TabFact | FetaQA |
|---|---|---|---|---|
| 1 | 1 | 61.60 | 88.34 | 35.68 |
| 3 | 3 | 66.11 | 90.42 | 35.93 |
| 5 | 5 | 66.92 | 90.81 | 37.15 |
| 1 | 3 | 64.46 | 85.42 | 35.98 |
| 3 | 1 | 61.14 | 89.18 | 35.73 |
| 1 | 5 | 69.38 | 82.16 | 36.17 |
| 5 | 1 | 71.04 | 87.35 | 36.03 |
| 3 | 5 | 66.62 | 86.51 | 35.99 |
| 5 | 3 | **76.38** | **91.30** | **36.34** |

We also observe that CIT-Agent consistently outperforms CIT-DP on WTQ and TabFact, likely due to its structured execution over tables and ability to handle large inputs with precise symbolic operations. On FetaQA, however, CIT-DP performs comparably, as BLEU evaluation favors the fluency of natural language outputs generated by DP, which better align with reference answers.

## 6.2 Effect of $n$ and $m$ in Answer Aggregation

To reduce bias from single-pass or single-mode inference, CIT performs $n$ rounds of Direct Prompting and $m$ rounds of Agent reasoning, aggregating all $n + m$ results via majority voting. As shown in Table 4, performance improves with larger $n$ or $m$. More Agent results tend to yield better accuracy, while DP offers complementary signals. These results underscore the value of reasoning diversity and validate the design of our causal aggregation framework.

## 6.3 Ablation Study

To assess the contribution of each component, we conduct ablation studies under both DP and Agent. We consider two variants: (1) *w/o Evidence Aggregation*, which uses evidence extracted from the original question $q$ for all variants; and (2) *w/o Question Variants*, which disables variant generation and reasons directly over $q$. Results in Table 5 show that removing evidence aggregation yields a minor drop, as most variants retrieve similar evidence. In contrast, removing question variants leads to a significant decline, confirming their importance in mitigating confounding bias. These findings validate the role of front-door variant generation in enabling robust, causally grounded inference.

Table 5: Ablation results on datasets.

| Method | WTQ | TabFact | FetaQA |
|---|---|---|---|
| **CIT-DP** | **66.40** | **83.15** | **33.15** |
| w/o Question Variants | 62.66 (↓ 3.74) | 76.38 (↓ 6.77) | 31.71 (↓ 1.44) |
| w/o Evidence Aggregation | 63.31 (↓ 3.09) | 79.64 (↓ 3.51) | 32.43 (↓ 0.72) |
| **CIT-Agent** | **73.76** | **90.61** | **36.15** |
| w/o Question Variants | 68.39 (↓ 5.37) | 86.86 (↓ 3.75) | 34.21 (↓ 1.94) |
| w/o Evidence Aggregation | 71.39 (↓ 2.37) | 87.99 (↓ 2.62) | 35.38 (↓ 0.77) |
| **CIT-DP&Agent** | **76.38** | **91.30** | **36.34** |
| w/o Question Variants | 71.50 (↓ 4.88) | 89.87 (↓ 1.43) | 35.41 (↓ 0.93) |
| w/o Evidence Aggregation | 74.47 (↓ 1.91) | 90.27 (↓ 1.03) | 36.02 (↓ 0.32) |

## 6.4 Generalization Across LLMs

We evaluate CIT on a diverse set of open-source and closed-source LLMs, using the same setup as Section 5.2. As shown in Table 6, CIT consistently improves performance across all models, regardless of size, architecture, or pretraining corpus. These results highlight the transferability of our causal intervention framework and suggest that core TableQA challenges—semantic ambiguity, spurious correlations, and evidence selection—are shared across LLMs.

Table 6: Comparison of LLMs with and without CIT.

| | Models | Init Accuracy(%) | + CIT Accuracy(%) |
|---|---|---|---|
| Open-source | Llama 2-7b | 48.34 | 49.95 (**1.61** ↑) |
| | Llama 2-13b | 50.18 | 52.07 (**1.89** ↑) |
| | Llama 2-70b | 59.02 | 61.33 (**2.31** ↑) |
| | DeepSeek R1 | 78.38 | 80.48 (**2.10** ↑) |
| Closed-source | GLM 4 | 65.84 | 66.53 (**0.69** ↑) |
| | GPT 4 | 70.89 | 77.09 (**6.20** ↑) |
| | Gemini 1.5 | 61.56 | 66.51 (**4.95** ↑) |
| | Claude 3.5 | 72.33 | 75.87 (**3.54** ↑) |

## 6.5 Analysis of Influential Factors

**Effect of Variant Quantity.** As shown in Figure 3, accuracy improves with more question variants due to greater semantic coverage, but saturates beyond three. Token usage also increases sharply, especially during reasoning. We set the default to three variants to balance performance and efficiency.

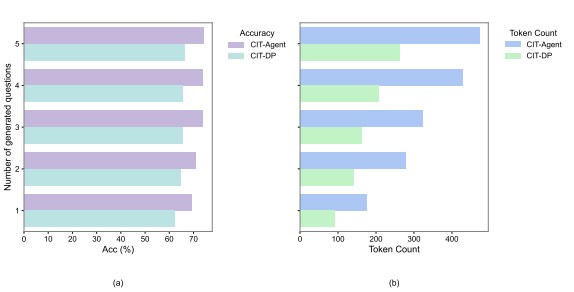
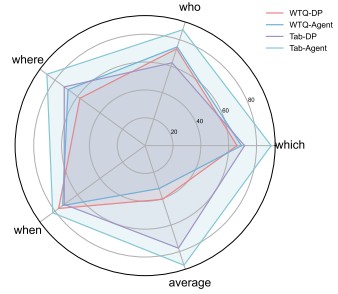

Figure 3: Changes in accuracy and token consumption under different numbers of question variants.

Figure 4: Impact of question type on CIT performance.

**Performance Across Question Types** We analyze CIT across question types on WTQ and TabFact (Figure 4), excluding FetaQA due to BLEU's incompatibility with discrete categories. Questions are grouped by keywords (e.g., *who*, *when*, *average*). On TabFact, both modes perform well across types. On WTQ, performance drops on numerical reasoning, especially aggregation. CIT-Agent further struggles due to hallucinated code constraints (e.g., unnecessary `unique()`). Despite this, the two modes show complementary strengths, supporting our joint reasoning.

**Impact of LLM Size.** We evaluate CIT using LLaMA 2 models of different sizes. As shown in Table 6, larger LLMs consistently yield better performance, likely due to their enhanced knowledge representation and reasoning capabilities. These strengths improve both the quality of question variants and the accuracy of evidence selection, which are critical to the effectiveness of front-door adjustment. This suggests that CIT benefits from scaling and can serve as a lightweight debiasing layer for increasingly powerful LLMs.

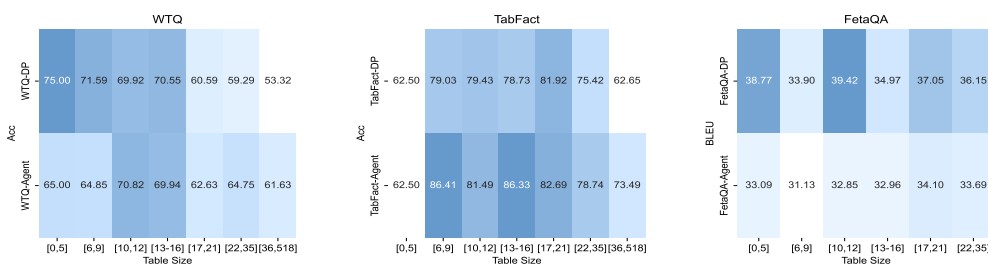

Figure 5: Impact of table size on TableQA performance.

**Table Size Sensitivity.** We evaluate CIT's robustness under varying table sizes by grouping test samples into bins of roughly 430 examples and computing average accuracy per bin (Figure 5). While accuracy generally decreases with larger tables, CIT remains stable overall. CIT-Agent outperforms CIT-DP on larger tables, as it executes Python code over full tables, bypassing context length limits, whereas DP's reliance on in-context reasoning leads to degraded performance with long inputs.

**Efficiency and API Usage.** CIT is efficient and compatible with both open-source and closed-source LLMs. As shown in Table 6, it performs well on non-API models like LLaMA and DeepSeek, supporting private deployment. For API-based use, CIT requires only three calls: one each for variant generation, evidence integration, and answer inference. Table 7 shows that CIT achieves strong performance with substantially fewer API calls than prior LLM-based methods.

Table 7: Comparison of methods with results and API calls

| Methods | Result | Number of API calls |
|---|---|---|
| CHAIN-OF-TABLE | 59.94 | (Next Operation 1 + Argument 1 + Transform 1) * Iter $N = 3N$ |
| CIT | 76.38 | Generate Questions 1 + Evidence Integration 1 + Answer 1 = 3 |

**Error Case Analysis.** We manually examine 100 examples to identify common sources of failure. For CIT-DP, most errors arise from incorrect answer formatting and the inability to recognize special table lines such as headers, footnotes, or merged cells. For CIT-Agent, errors often involve hallucinated constraints—such as adding non-existent conditions—and occasional format inconsistencies. These findings highlight the challenges of aligning LLM output with task-specific answer expectations. A detailed breakdown of error types and representative examples is provided in Appendix F.

**Additional Analyses.** We further investigate whether CIT is affected by LLM data contamination, details are reported in Appendix E. We also enumerate the range of question types that CIT can handle effectively—including *where*, *when*, *which*, *what*, *who*, *is/does*, *how many*, *average*, *sum*, etc.—with corresponding case examples shown in Appendix G.

# 7    Limitations

While CIT offers a principled way to mitigate confounding bias, its effectiveness depends on the quality of question variants. Limited diversity or semantic inconsistency may hinder coverage of the original intent. Future work may explore controlled generation or filtering to improve variant quality.

# 8    Conclusion

We present CIT, a causal intervention framework for TableQA that applies front-door adjustment to mitigate latent confounding bias in LLM-based reasoning. By modeling TableQA as a structural causal process, CIT identifies and blocks spurious back-door paths introduced by pretraining. The method implements this via question variant generation, evidence aggregation, and joint reasoning with both Direct Prompting and Agent paradigms. Extensive results across multiple benchmarks and LLMs demonstrate CIT's robustness, effectiveness, and generality.

# 9    Acknowledgements

Our work is supported by the National Natural Science Foundation of China (62476003), Anhui Province Excellent Scientific Research and Innovation Team (2024AH010004), Anhui Provincial Natural Science Foundation - Water Science Joint Fund (2408055US006), the University Synergy Innovation Program of Anhui Province (GXXT-2023-050), and SMP-Zhipu.AI Large Model Cross-Disciplinary Fund (SMP-Zhipu20240210). We also acknowledge the support from Zhipu AI-Anhui University Joint Research Center, and the High-Performance Computing Platform of Anhui University.

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

## A  Related Work: LLM-based TableQA

**Direct Prompting in TableQA.**   In Direct Prompting (DP), LLMs perform step-by-step reasoning through serialized natural language prompts, often under the Chain-of-Thought (CoT) framework [Kong et al., 2024, Zhao et al., 2023b, Deng et al., 2024]. Early DP methods [Zhao et al., 2023a, Sui et al., 2023, Chemmengath et al., 2021] used few-shot examples, SQL-style prompts, or zero-shot CoT to help LLMs decompose and solve complex queries. For instance, Luo et al.[Luo et al., 2023] constructed CoT exemplars with retrieval-based reconstruction, while BINDER[Cheng et al., 2023] composed sub-queries via SQL logic. DATER [Ye et al., 2023b] guided reasoning through SQL-based parsing and completion. More recently, [Liu et al., 2024a] explored zero-shot prompting with "think step by step" instructions to encourage implicit decomposition.

**Agent in TableQA.**   In the Agent paradigm, LLMs analyze the question, plan steps, and generate Python code to operate over tables [Li et al., 2024a, Gong et al., 2020]. CHAIN-OF-TABLE [Wang et al., 2024] decomposes questions by creating intermediate tables and applying custom functions. ReAcTable [Zhang et al., 2024c] iteratively generates intermediate results and adapts subsequent actions based on output. Other works [Liu et al., 2024b,a] integrate SQL or Python-based agents for structured code-level reasoning.

**Joint DP and Agent.**   DP and Agent can be combined for joint reasoning. Mix-SC [Liu et al., 2024a] merges both paradigms and uses majority voting for answer selection. TIDE [Yang et al., 2025] further introduces structured triplets to enhance decomposition. However, current methods focus on guiding reasoning without addressing token co-occurrence bias from LLM pretraining, which can introduce spurious correlations. To address this, we are the first to define LLM-based TableQA from a causal perspective and identify latent confounding in the reasoning process.

## B  Front-door Adjustment

Back-door adjustment requires explicit access to the confounding variable $C$. However, in our setting, the bias induced by LLM pretraining is latent and unobservable, rendering back-door adjustment inapplicable. Fortunately, the front-door adjustment criterion [Pearl et al., 2016] enables causal estimation without requiring access to confounder values. It operates by intervening on the treatment variable using the *do*-operator and leveraging an observed mediator that satisfies the front-door conditions.

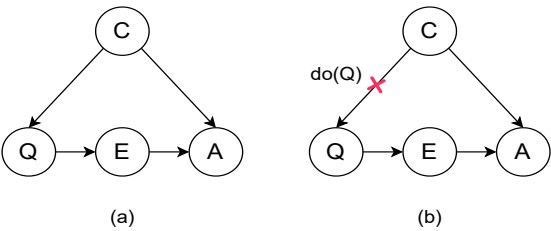

Figure 6: The causal graph of TableQA.

Following the law of total probability, we derive the decomposition expressed in Equation 9.

$$P(A = a \mid do(Q = q)) = \sum_e P(A = a \mid do(Q = q), E = e)P(E = e \mid do(Q = q)) \quad (9)$$

By the back-door criterion, the intervention on $E$ does not alter the conditional distribution of $A$ given $Q$ and $E$. Hence, introducing the *do*-operator on $E$ does not affect the overall expression, yielding Equation 10.

$$P(A = a \mid do(Q = q)) = \sum_e P(A = a \mid do(Q = q), do(E = e))P(E = e \mid do(Q = q)) \quad (10)$$

Under the structural causal model (SCM), since $Q$ and $E$ are connected via a direct causal link without confounders, the *do*-operator on $Q$ can be omitted, yielding Equation 11.

$$P(A = a \mid do(Q = q)) = \sum_e P(A = a \mid do(Q = q), do(E = e))P(E = e \mid Q = q) \quad (11)$$

In the SCM, Q and A are not directly connected, so intervening on Q does not affect the distribution of A, yielding Equation 12.

$$P(A = a \mid do(Q = q)) = \sum_e P(A = a \mid do(E = e))P(E = e \mid Q = q) \quad (12)$$

Using the law of total probability, we can derive Equation 13.

$$P(A = a \mid do(Q = q)) = \sum_{q'} \sum_e P(A = a \mid do(E = e), Q = q')P(Q = q' \mid do(E = e))P(E = e \mid Q = q) \quad (13)$$

Using the same logic as in the transition from Equation 9 to Equation 10, we proceed as Equation 14.

$$P(A = a \mid do(Q = q)) = \sum_{q'} \sum_e P(A = a \mid E = e, Q = q')P(Q = q' \mid do(E = e))P(E = e \mid Q = q)$$
$$(14)$$

Using the same logic as in the transition from Equation 10 to Equation 11, we proceed as Equation 15.

$$P(A = a \mid do(Q = q)) = \sum_{q'} \sum_e P(A = a \mid E = e, Q = q')P(Q = q')P(E = e \mid Q = q) \quad (15)$$

Finally, by reorganizing the summation terms, we obtain the Equation 16.

$$P(A = a \mid do(Q = q)) = \sum_e P(E = e \mid Q = q) \sum_{q'} P(A = a \mid E = e, Q = q')P(Q = q') \quad (16)$$

## C    Prompt

We provide the prompts for the three core components in this section.

## D    Baselines

SASP [Ou and Liu, 2022] uses lexical and structural features to generate programs for solving pseudo programs. TAPAS-large [Eisenschlos et al., 2020] creates a balanced dataset of millions of automatically generated training examples for intermediate learning before fine-tuning. T5-3B [Xie et al., 2022] within the Unified SKG framework unifies 21 SKG tasks into a text-to-text format for comprehensive SKG research. TAPEX-large [Liu et al., 2021] learns a neural SQL executor on a synthetic corpus of executable SQL queries and their outputs. Task Configs [Chen et al., 2023] structured compositional task prompts improve multi-task learning and zero-shot generalization for table-to-text models. TARGET [Ji et al., 2024] is a benchmark for table retrieval in generative tasks, evaluating retriever performance and downstream impacts on QA, fact-checking, and text-to-SQL. TabCot [Chen, 2023b] LLMs excel at table reasoning via chain-of-thought prompting, matching specialized models without table-specific training. TAG-QA [Zhao et al., 2023a] pioneers graph-guided + knowledge-augmented TableQA for long-form answers. UniTabPT[Sarkar and Lausen, 2023] Unified table-pretrained LLMs (T5-based) that outperform specialized models across parsing/QA/classification at scale (770M–11B).

Codex [Cheng et al., 2023], as an OpenAI API, can generate SQL or Python statements and perform end-to-end QA. BINDER [Cheng et al., 2023] combines end-to-end and symbolic approaches,

**Instruction:** Based on the table content, generate a question similar to the original question without changing its main content, and write it after 'generate questions:'.

**Example:**
Table:
/*
table caption : stay in office
| name | took office | left office | party |
|---:|-------:|:-------------------|:-------|:----|----------------------:|
| William McCreery | March 4, 1803 | March 3, 1809 | Democratic Republican |
| Alexander McKim | March 4, 1809 | March 3, 1815 | Democratic Republican |
| William Pinkney | March 4, 1815 | April 18, 1816 | Democratic Republican |
| Peter Little | September 2, 1816 | March 3, 1823 | Democratic Republican |
| Peter Little | March 4, 1823 | March 3, 1825 | Jacksonian DR |
| Peter Little | March 4, 1825 | March 3, 1829 | Adams |
| Benjamin C. Howard | March 4, 1829 | March 3, 1833 | Jacksonian |
*/
Question: How many people stayed at least 3 years in office?
generate questions:
1. The total number of people stay at least 3 years in office?
2. How many people stayed more than 3 years in office?
3. How many people served in office for 3 years or more?

**Test:**
Table:
/*
table caption : {TITLE}
{TABLE}
*/
Question: {QUESTION}

Figure 7: The prompt of question variants generation.

**Instruction:** Analyze the initial questions and the generated similar questions based on the table, find the relevant evidence related to each question, and summarize the questions to obtain the total evidence, and write it after ' evidence:'.

**Test:**
Table:
/*
table caption : {TITLE}
{TABLE}
*/
Question: {QUESTION}

Figure 8: The prompt of evidence aggregation.

generating and iteratively refining pseudo-SQL queries to construct final answers. For TableQA, DATER [Ye et al., 2023b] extracts relevant sub-tables and decomposes questions to reason jointly over them. StructGPT [Jiang et al., 2023] enhances zero-shot reasoning by iterating through specialized interfaces for structured data. DTE [Wang et al., 2023a] generates counterfactual examples to refine text-to-SQL question answering. TACR [Wu et al., 2023] aligns multi-hop questions with different modalities for accurate evidence retrieval. ITR [Lin et al., 2023] selects relevant rows and columns to form a compact sub-table for efficient reasoning.

[Liu et al., 2024b] creates new tables with external information, enabling SQL queries over both original and new tables to answer. CHAIN-OF-TABLE [Wang et al., 2024] dynamically plans operation chains based on table structure and associated questions. ReAcTable [Zhang et al., 2024c] uses LLMs to iteratively generate intermediate tables, with external code execution for accuracy. Cabinet [Patnaik et al., 2024] removes irrelevant noise in tables to improve LLM reasoning accuracy. Mix-SC [Liu et al., 2024a] explores the combination of CoT and PyAgent to address LLM sensitivity to table structure. SYNTQA [Zhang et al., 2024b] unifies Text-to-SQL (arithmetic/long tables) and

Figure 9: The prompt of answer inference.

E2E TQA (ambiguity/schemas) via answer selection, boosting performance. TIDE [Yang et al., 2025] use structuring triples to help LLMs decompose and validation reasoning context.

# E  Data Contamination

**Mitigating Data Contamination with CIT.** Data contamination is a common concern in LLM-based methods, where test samples may appear in the model's training data. To evaluate the robustness of CIT, we compare the performance of direct answering with that of CIT-based reasoning. As shown in Table 8, CIT achieves approximately 27% higher accuracy, indicating that its effectiveness primarily stems from the method itself rather than potential data leakage.

Table 8: Comparison of direct QA for data contamination.

| Models | Accuracy(%) |
|-------|---------|
| Direct QA [Cheng et al., 2023] | 48.70 |
| **CIT** | **76.38** |

| # | Description | 1939/40 | 1940/41 | 1941/42 | 1942/43 | 1943/44 | 1944/45 | Total |
|---|---|---|---|---|---|---|---|---|
| 0 | Direct War Losses | 360,000 | NaN | NaN | NaN | NaN | 183,000 | 543,000 |
| 1 | Murdered | 75,000 | 100,000 | 116,000 | 133,000 | 82,000 | NaN | 506,000 |
| 2 | Deaths In Prisons & Camps | 69,000 | 210,000 | 220,000 | 266,000 | 381,000 | NaN | 1,146,000 |
| 3 | Deaths Outside of Prisons & Camps | NaN | 42,000 | 71,000 | 142,000 | 218,000 | NaN | 473,000 |
| 4 | Murdered in Eastern Regions | NaN | NaN | NaN | NaN | NaN | 100,000 | 100,000 |
| 5 | Deaths in Other Countries | NaN | NaN | NaN | NaN | NaN | NaN | 2,000 |
| 6 | **Total** | 504,000 | 352,000 | 407,000 | 541,000 | 681,000 | 270,000 | 2,770,000 |

| | |
|---|---|
| Question : | how many people were murdered in 1940/41? |
| Gold Answer : | 100,000 |
| Reason Answer : | 100000 |
| Error Analysis : | **Incorrect answer format** |

| | |
|---|---|
| Question : | what is the last description of losses on this chart? |
| Gold Answer : | Deaths other countries |
| Reason Answer : | Total |
| Error Analysis : | **Unable to recognize special line (total)** |

Figure 10: Incorrect answer format and unable to recognize special line errors.

| Date | Competition | Location | Country | Event | Placing | Nationality |
|---|---|---|---|---|---|---|
| 31 October 2008 | 2008–09 World Cup | Manchester | United Kingdom | Keirin | 2 | GBR |
| 31 October 2008 | 2008–09 World Cup | Manchester | United Kingdom | Sprint | 1 | GBR |
| 1 November 2008 | 2008–09 World Cup | Manchester | United Kingdom | 500 m time trial | 1 | GBR |
| 1 November 2008 | 2008–09 World Cup | Manchester | United Kingdom | Sprint | 1 | GBR |
| 2 November 2008 | 2008–09 World Cup | Manchester | United Kingdom | Team sprint | 1 | GBR |
| 2 November 2008 | 5th International | Manchester | United Kingdom | International keirin | 2 | GBR |
| 2 November 2008 | 2008–09 World Cup | Manchester | United Kingdom | Team sprint | 1 | GBR |
| 2 November 2008 | 2008–09 World Cup | Manchester | United Kingdom | Keirin | 1 | GBR |
| 2 November 2008 | 2008–09 World Cup | Manchester | United Kingdom | Team sprint | 1 | GBR |
| 13 February 2009 | 2008–09 World Cup | Copenhagen | Denmark | Team sprint | 1 | GBR |
| 13 February 2009 | 2008–09 World Cup | Copenhagen | Denmark | Team sprint | 1 | GBR |
| 13 February 2009 | 2008–09 World Cup | Copenhagen | Denmark | Team sprint | 1 | GBR |
| 13 February 2009 | 2008–09 World Cup | Copenhagen | Denmark | Sprint | 1 | GBR |
| 30 October 2009 | 2009–10 World Cup | Manchester | United Kingdom | Sprint | 1 | GBR |
| 30 October 2009 | 2009–10 World Cup | Manchester | United Kingdom | Sprint | 1 | GBR |
| 30 October 2009 | 2009–10 World Cup | Manchester | United Kingdom | Keirin | 1 | GBR |
| 30 October 2009 | 2009–10 World Cup | Manchester | United Kingdom | 500 m time trial | 2 | GBR |
| 1 November 2009 | 2009–10 World Cup | Manchester | United Kingdom | Team sprint | 1 | GBR |
| 1 November 2009 | 2009–10 World Cup | Manchester | United Kingdom | Team sprint | 1 | GBR |
| 1 November 2009 | 2009–10 World Cup | Manchester | United Kingdom | Team sprint | 1 | GBR |

| | |
|---|---|
| Question : | what is the total number of competition? |
| Gold Answer : | 20 |
| Reason Answer : | unique_competitions = df['Competition'].unique(), Final answer is 3. |
| Error Analysis : | **Hallucination adds extra conditions (unique)** |

Figure 11: Halluciantion adds extra conditions error.

| Rank | Cyclist | Team | Time | UCI ProTour Points |
|---|---|---|---|---|
| 1 | Alejandro Valverde (ESP) | Caisse d'Epargne | 5h 29' 10" | 40 |
| 2 | Alexandr Kolobnev (RUS) | Team CSC Saxo Bank | s.t. | 30 |
| 3 | Davide Rebellin (ITA) | Gerolsteiner | s.t. | 25 |
| 4 | Paolo Bettini (ITA) | Quick Step | s.t. | 20 |
| 5 | Franco Pellizotti (ITA) | Liquigas | s.t. | 15 |
| 6 | Denis Menchov (RUS) | Rabobank | s.t. | 11 |
| 7 | Samuel Sánchez (ESP) | Euskaltel-Euskadi | s.t. | 7 |
| 8 | Stéphane Goubert (FRA) | Ag2r-La Mondiale | + 2" | 5 |
| 9 | Haimar Zubeldia (ESP) | Euskaltel-Euskadi | + 2" | 3 |
| 10 | David Moncoutié (FRA) | Cofidis | + 2" | 1 |

| | |
|---|---|
| Question : | who was ranked next after davide rebellin? |
| Gold Answer : | Paolo Bettini (ITA) |
| Reason Answer : | Paolo Bettini |
| Error Analysis : | **Incorrect answer format** |

Figure 12: Incorrect answer format error.

# F   Error Case

# G   Question Types

| | |
|---|---|
| Question : | **what** country had the most cyclists? |
| Question Variant : | 1.  which country had the highest number of cyclists? 
 2.  which nation had the largest representation of cyclists? 
 3.  in which country were the most cyclists from? |

| | |
|---|---|
| Question : | **who** was ranked between denis menchov and stephane goubert? |
| Question Variant : | 1.  which cyclist was positioned between denis menchov and stéphane goubert? 
 2.  who ranked just ahead of stéphane goubert and just behind denis menchov? 
 3.  who was the cyclist that came in between denis menchov and stéphane goubert in the rankings? |

| | |
|---|---|
| Question : | **which** year has the most deaths outside of prisons & camps? |
| Question Variant : | 1.  in which year were there the highest number of deaths outside of prisons & camps? 
 2.  which year had the greatest number of deaths outside of prisons & camps? 
 3.  in which year were the deaths outside of prisons & camps the highest? |

| | |
|---|---|
| Question : | **where** was the last competition played? |
| Question Variant : | 1.  what was the location of the final competition? 
 2.  in which venue did the last competition take place? 
 3.  where did hannes hopley compete last? |

Figure 13: Type of wh- questions.

| Question : | **how long** did it take the industrial quest to complete the course? |
|---|---|
| Question Variant : | 1. what was the elapsed time for the industrial quest to finish the race? 
 2. how long did the industrial quest take to complete the sydney to hobart yacht race? 
 3. what was the total time taken by the industrial quest to finish the course? |

| Question : | **how many** total films have they already appeared in? |
|---|---|
| Question Variant : | 1. what is the total number of films they have appeared in? 
 2. how many films has radhika pandit featured in? 
 3. what is the count of films that radhika pandit has acted in? |

| Question : | in cycle 4 of austria's next top model, what is the **average** of all the contestants' ages? |
|---|---|
| Question Variant : | 1. what is the average age of all the contestants in cycle 4 of austria's next topmodel? 
 2. what is the mean age of the contestants in cycle 4 of austria's next topmodel? 
 3. what is the average age of participants in the fourth cycle of austria's next topmodel? |

Figure 14: Type of how- and average questions.

