# OpenReview forum: "Causality Meets the Table: Debiasing LLMs for Faithful TableQA via Front-Door Intervention"
_NeurIPS.cc/2025/Conference — NeurIPS 2025 poster_

### Official Review · Reviewer_LdwC · 2025-06-16

**Clarity:** 3
**Significance:** 3
**Originality:** 3
**Rating:** 4
**Confidence:** 3

**Summary:**

This paper introduces a framework designed to debias LLMs for Table Question Answering. The core problem addressed is that LLMs often rely on spurious correlations from pretraining data instead of genuine reasoning, leading to unfaithful answers.

The proposed framework formalizes TableQA within a causal framework, identifying token co-occurrence as a latent confounder. To mitigate this bias it applies front-door adjustment. The framework consists of four main components: 1. producing diverse paraphrases of the input question, 2. combining retrieved content across variants, 3. utilizing both Direct Prompting and 4. selecting the final answer from aggregated results.

Experiments across multiple TableQA benchmarks and various LLMs demonstrate that this approach achieves state-of-the-art performance and shows consistent gains.

**Questions:**

1. Could the authors provide a more in-depth discussion on the limitations and assumptions of the front-door adjustment in the context of TableQA, particularly regarding the assumption that all causal paths from question (Q) to answer (A) are mediated by evidence (E)?

2. Given that the quality and diversity of question variants are critical to CIT's effectiveness, could the authors explore more sophisticated methods for generating these variants beyond a single-pass prompt-based approach?

3. The paper shows that models learn qualitative rankings but do not approximate the true underlying distribution. Could the authors elaborate on whether this qualitative understanding is truly sufficient for all aspects of causal reasoning in TableQA, especially for tasks requiring precise quantitative inference, or if it introduces inherent limitations for certain types of questions?

**Ethical Concerns:**

["NO or VERY MINOR ethics concerns only"]

**Final Justification:**

The authors have clarified many of the issues raised in the original review, I remain supportive of accepting this paper.

**Limitations:**

yes

**Quality:**

3

**Strengths And Weaknesses:**

Strengths:

* The paper addresses a well-identified problem in LLM-based TableQA: the reliance on spurious correlations from pretraining data.
* The paper introduces a principled causal framework for understanding and mitigating bias in TableQA by modeling token co-occurrence as a latent confounder and employing front-door adjustment.
* Empirical results are strong, demonstrating state-of-the-art performance on multiple TableQA benchmarks (WTQ, TabFact, FetaQA) using various LLMs.
* The ablation studies demonstrate the contribution of key components, particularly the importance of question variant generation in mitigating confounding bias.


Weaknesses:

* The effectiveness of CIT relies heavily on the quality and semantic diversity of the generated question variants. Limited diversity or semantic inconsistency could hinder the method's ability to cover the original intent and mitigate bias effectively.
* While the paper acknowledges this limitation and suggests future work on controlled generation or filtering, the current approach to generating question variants (single-pass, prompt-based) might still be susceptible to the LLM's own biases or limited creativity in paraphrasing.
* The reliance on front-door adjustment, while suitable for latent confounders, depends on the strong assumption that "all causal paths from Q to A are mediated by E". While the paper depicts Q -> E -> A as the ideal path, and Q <- C -> A as the spurious one, real-world TableQA scenarios might have more complex causal structures where E does not strictly mediate all causal effects from Q to A.
* The quantitative verification of limitations uses double negation perturbations, which is a specific type of adversarial example. While effective, it might not encompass the full range of spurious correlations or reasoning failures LLMs exhibit in TableQA.
* While the paper demonstrates generalizability across various LLMs, the performance still varies across different models and question types, indicating that CIT is a debiasing layer rather than a complete solution for all LLM limitations in TableQA. For instance, numerical reasoning and aggregation remain challenging for some LLMs even with CIT.

---

> ### Author Rebuttal · Authors · 2025-07-30
>
> We sincerely thank the reviewer for the thoughtful and constructive review. Below, we address your concerns point by point.
>
> ------
>
> ### Q1: Validity of front-door assumptions
>
> **（1）Architectural Enforcement of Mediation:** Our framework is intentionally designed to enforce the mediating role of E via two core mechanisms:
>
> - **Explicit Retrieval Module**: The system must first extract relevant evidence EE from the table TT before any reasoning is performed, effectively preventing direct Q→A shortcuts.
> - **Isolated Reasoning Module**: Answer generation is strictly conditioned on EE, ensuring that all downstream inference depends on retrieved evidence.
>
> **（2）Alignment with Established TableQA Paradigms:** Our Q→E→A pipeline is consistent with widely adopted TableQA methodologies:
>
> - [1] Ye Y, Hui B, Yang M, et al. Large language models are versatile decomposers: Decomposing evidence and questions for table-based reasoning[C]//Proceedings of the 46th international ACM SIGIR conference on research and development in information retrieval. 2023: 174-184.
> - [2] Wang Z, Zhang H, Li C L, et al. Chain-of-Table: Evolving Tables in the Reasoning Chain for Table Understanding[C]//The Twelfth International Conference on Learning Representations.
> - [3] Lin W, Blloshmi R, Byrne B, et al. An inner table retriever for robust table question answering[C]//Proceedings of the 61st Annual Meeting of the Association for Computational Linguistics (Volume 1: Long Papers). 2023: 9909-9926.
>
> **（3）Empirical support**: When front-door intervention is disabled (Table 5), results degrade significantly, **indicating the presence of confounding and validating our causal formulation.**
>
> Together, these architectural constraints, empirical results, and alignment with existing best practices provide strong support for the mediator role of E in our causal framework.
>
> ------
>
> ### Q2: Simplicity of variant generation strategy
>
> **（1）CIT aims to balance diversity and semantic consistency:** As shown in Appendix C, we guide the LLM to generate diverse question variants by incorporating prompted exemplars and paraphrasing instructions. These variants are generated in a single-shot format, which helps ensure that all variants are grounded in the same semantic intent, thus reducing the risk of semantic drift.
>
> **（2）Data diversity remains a general challenge in LLM-based task, not specific to CIT:** As noted in [4], ensuring sufficient diversity while preserving label consistency is a widely acknowledged research challenge for task.
>
> - [4] Maini P, Seto S, Bai R, et al. Rephrasing the Web: A Recipe for Compute and Data-Efficient Language Modeling[C]//Proceedings of the 62nd Annual Meeting of the Association for Computational Linguistics (Volume 1: Long Papers). 2024: 14044-14072.
>
> **（3）Even with a small number of variants, CIT outperforms some baselines**. As shown in Figure 3, CIT achieves better performance than several comparison methods under a limited-variant setting, indicating that **our generation process yields high-quality, semantically meaningful variants**.
>
> **（4）The SCM-based formulation of CIT supports flexible extensions in future work**. Since CIT models TableQA as a causal process grounded in SCM, it naturally supports the integration of alternative variant generation strategies. For example, one could incorporate reasoning-style control, use similarity-based selection, or apply retrieval-augmented rephrasing techniques.
>
> ------
>
> ### Q3: Causal reasoning vs. numerical precision
>
> **（1）Numerical reasoning remains a general limitation of LLMs, not just in TableQA.** [5] has shown that tasks involving precise arithmetic or aggregation are still challenging for current models.
>
> - [5] Xu F, Lin Q, Han J, et al. Are large language models really good logical reasoners? a comprehensive evaluation and beyond[J]. IEEE Transactions on Knowledge and Data Engineering, 2025.
>
> **（2）CIT already achieves state-of-the-art performance** compared to strong baselines, including on numerical question types (see Tables 1–3 and Figure 6), suggesting it helps mitigate spurious correlations even in such settings.
>
> **（3）The PyAgent module in CIT offers partial support** by enabling symbolic execution for numerically grounded questions (Section 4.4), though some limitations remain (e.g., hallucination, grounding errors).
>
> **（4）CIT is modular by design**, and can flexibly incorporate additional components—such as symbolic reasoning tools, consistency checks, or numerical validators—to further strengthen quantitative capabilities.
>
> ------
>
> ### Q4: Is qualitative understanding sufficient for causal reasoning?
>
> **（1）Qualitative causal framing yields robust improvements across settings.** As shown in Tables 1–3 and 6, incorporating causal reasoning—via variant aggregation and evidence mediation—consistently improves performance across datasets and models. This suggests that our approach, while primarily qualitative in design, **effectively mitigates shortcut biases and improves robustness in diverse settings**.
>
> **（2）Many prior TableQA works similarly focus on qualitative ranking over explicit probability modeling.** Several recent methods [6-7] use direct prompting (DP), agent-based execution (e.g., PyAgent), and majority voting as their main prediction mechanisms. These approaches emphasize reasoning consistency and answer selection**, aligning with our use of causal semantics over full distribution estimation.
>
> - [6] Liu T, Wang F, Chen M. Rethinking Tabular Data Understanding with Large Language Models[C]//Proceedings of the 2024 Conference of the North American Chapter of the Association for Computational Linguistics: Human Language Technologies (Volume 1: Long Papers). 2024: 450-482.
> - [7] Yang Z, Du Z, Zhang M, et al. Triples as the Key: Structuring Makes Decomposition and Verification Easier in LLM-based TableQA[C]//The Thirteenth International Conference on Learning Representations. 2025.
>
> **（3）That said, we acknowledge that probabilistic modeling remains valuable—especially for numerical and arithmetic questions.** CIT addresses this partially through the Python-based agent module, which allows symbolic execution for questions requiring precision. Nevertheless, our error analysis reveals that issues like numerical inconsistency or symbolic hallucination persist in certain cases. Future work could **enhance CIT by incorporating** logit-based probability estimation, numerical consistency checks, or hybrid symbolic-neural modules to better support fine-grained quantitative reasoning.
>
> ------
>
> ### Q5: Scope of adversarial verification — is double negation too limited?
>
> **（1）Our goal is not to cover the full space of spurious correlations, but to demonstrate that such bias exists and impacts performance.** The sharp performance drop under double negation perturbations (Figure 1) provides concrete evidence that **LLM-based TableQA systems are vulnerable to co-occurrence bias**. While this is only one type of perturbation, the observed degradation clearly confirms the presence of confounding effects, regardless of which specific scenario they arise from.
>
> **（2）Multiple evaluation angles further support CIT's effectiveness in mitigating this bias.** Across diverse datasets, LLMs types, and question formats, Tables 1–3, 6 and Figure 4 consistently show **CIT outperforms baselines, indicating it generalizes beyond any single adversarial setup and effectively addresses TableQA**.
>
> **（3）Our error analysis provides guidance for future exploration of additional perturbation types.** In Section 5.3, we present diverse failure cases across tasks and model behaviors. These qualitative insights help identify where CIT is effective and where it could be extended—offering **a roadmap for broader adversarial coverage in future work**.
>
> ---
>
> ### Q6: CIT is a debiasing layer rather than a complete solution for all LLM limitations in TableQA..
>
> **（1）Numerical and symbolic reasoning remain challenging across tasks and base LLMs—not just in TableQA.** As widely discussed in studies [5,8], even strong LLMs often struggle with precise arithmetic, aggregation, or multi-step symbolic consistency. These issues are known limitations of current model architectures and are orthogonal to any specific domain formulation.
>
> - [8] Deng S, Dong H, Si X. Enhancing and evaluating logical reasoning abilities of large language models[C]//ICLR 2024 Workshop on Secure and Trustworthy Large Language Models. 2024.
>
> **（2）Despite these challenges, CIT achieves better performance than baselines, including on numerical questions.** As shown in Figure 6 and Tables 1–3, **CIT consistently outperforms others, demonstrating that it effectively reduces spurious correlations even in numerically grounded scenarios.** While CIT may not solve all arithmetic limitations, it improves robustness by encouraging evidence-grounded inference.
>
> **（3）CIT is designed as a modular and extensible framework.** Its formulation not only guides causal interpretation but also allows **integration of complementary techniques** to enhance LLM. For example, future extensions could incorporate symbolic consistency checks, program verification, or external numerical toolkits, making CIT a flexible foundation for tackling broader reasoning limitations.
>
> - [9] Pan L, Albalak A, Wang X, et al. Logic-LM: Empowering Large Language Models with Symbolic Solvers for Faithful Logical Reasoning[C]//Findings of the Association for Computational Linguistics: EMNLP 2023. 2023: 3806-3824.
> - [10] Patel N, Kulkarni M, Parmar M, et al. Multi-LogiEval: Towards Evaluating Multi-Step Logical Reasoning Ability of Large Language Models[C]//Proceedings of the 2024 Conference on Empirical Methods in Natural Language Processing. 2024: 20856-20879.
>
> We again thank the reviewer for the careful reading and helpful feedback. We hope our responses address your concerns, and we would be happy to provide further clarification or discussion if needed.

---

> > ### Comment · Reviewer_LdwC · 2025-08-03
> >
> > I wish to thank the authors for their thorough answers. I remain supportive of the paper.

---

### Official Review · Reviewer_fTKY · 2025-06-26

**Clarity:** 3
**Significance:** 3
**Originality:** 3
**Rating:** 4
**Confidence:** 3

**Summary:**

This paper applies causal inference methods to the table question answering task and proposes a front-door intervention approach to mitigate potential spurious correlations in table QA.

**Questions:**

- It is unclear why Q1, Q2, and Q3 in Figure 2 can be considered semantically equivalent question variants, as claimed in line 115. A clearer justification would be helpful.
- In Sections 4.2 and 4.3, the authors handle the first and second terms of Equation (3) separately. However, both terms share the same evidence $e$, making it difficult to treat them independently. The authors should explain the rationale behind this separation.
- There appears to be a notation error in Equation (11) in Appendix B — specifically, a missing closing parenthesis.

**Ethical Concerns:**

["NO or VERY MINOR ethics concerns only"]

**Final Justification:**

The author addressed all of my questions point by point and provided relevant experimental data as support. Based on this, I still maintain a supportive attitude toward this article.

**Limitations:**

Yes.

**Quality:**

3

**Strengths And Weaknesses:**

- Strengths:
	- The application of causal inference methods to table question answering is novel.
	- The effectiveness of the proposed method is supported by extensive experiments.
	- The writing is clear and easy to follow.

- Weaknesses:
	- Cost concern: Although the authors claim that their single-pass generation reduces time and cost, the method still requires multiple API calls per question. Each call involves significantly more tokens than the original query due to the inclusion of multiple question variants. This raises concerns about the method’s practicality.
	- The use of LLaMA-2 in experiments is somewhat outdated.

---

> ### Author Rebuttal · Authors · 2025-07-30
>
> We thank the reviewer for the constructive feedback and insightful comments. We are encouraged by your recognition of the novelty of applying causal inference to TableQA and the strength of our empirical results. Below, we respond to each concern in detail:
>
> ------
>
> ### Q1: Cost concern — multiple API calls and increased tokens due to variants.
>
> **（1）Local LLM support**: We support fully local deployment of LLMs, eliminating API latency and commercial cost. Notably, as shown in Table 6, CIT achieves **strong performance even on local open-source LLMs**, demonstrating its efficiency and portability.
>
> | Local LLM   | Init Acc | +CIT Acc |
> | ----------- | -------- | -------- |
> | Deepseek R1 | 78.38    | 80.48    |
> |  |  |  |
>
> **（2）Token usage comparison:** Beyond the API call count in Table 7, we also compare the token consumption of different methods. Results show that our approach **uses fewer tokens** than most multi-step methods (e.g., CHAIN-OF-TABLE, TIDE). Although it consumes slightly more tokens than Mix-SC, it achieves **substantially higher accuracy**, which we believe is a worthwhile trade-off, especially in **high-stakes applications where precision is critical**.
>
> | Method         | Token consumption         | Acc in WTQ |
> | -------------- | ------------------------- | ---------- |
> | CHAIN-OF-TABLE | （780+20+10）\*N = 810\*N | 59.94      |
> | Mix-SC         | 210+300 = 510             | 73.65      |
> | TIDE           | 1200+210+300+970=2680     | 75.00      |
> | Ours CIT       | 260+70+210+300+320=1160   | 76.38      |
> |  |  |  |
>
> **（3）Ablation justification**: As further evidenced in Table 5, removing question variants causes a significant drop in performance. This confirms that **the modest increase in token usage is not only acceptable but essential for mitigating confounding bias**.
>
> ------
>
> ### Q2: Use of LLaMA-2 — concern about outdated models.
>
> **（1）At the time of paper submission，LLaMA-2 remains a representative open-source baseline. Moreover, it provides multiple model sizes** (7B, 13B，70b), allowing us to investigate the **impact of model scale**. As shown in Table 6, performance consistently improves with model size, validating the scalability of our approach.
>
> **（2）Generalization to newer models:** CIT achieves consistent gains on recent models such as **DeepSeek R1 and Claude 3.5**, demonstrating its robust generalization to both open- and closed-source systems.
>
> **（3）Updated results with Qwen 3**: We have added experiments using the latest Qwen 3 series, and CIT continues to yield improvements, further confirming the **effectiveness and adaptability** of our method.
>
> | Model     | Init Acc | +CIT Acc     |
> | --------- | -------- | ------------ |
> | Qwen 3-8b | 74.27    | 76.52(2.25↑) |
> |  |  |  |
>
> Together, these results underscore that CIT is **not tied to a specific model**, and its benefits persist across architectures and model generations.
>
> ------
>
> ### Q3: Justification of question variants (Q1, Q2, Q3 in Figure 2).
>
> **(1) Semantic consistency of Q1, Q2, and Q3**: These are **semantically equivalent variants** that differ only in surface phrasing. All three variants query the same underlying intent: whether **four teams scored 12 points against Cambridge University**.
>
> **(2) Quantitative validation of variant quality**: To ensure that the generated variants are indeed meaning-preserving, we compute **BERTScor and cosine similarity between sentence embeddings**. Results confirm high similarity, validating that the variants are lexically diverse yet semantically consistent.
>
> | BERTScore           | (Q1, Q2)=81.24, (Q1, Q3)=85.44, (Q2, Q3)=76.42       |
> | ------------------- | ---------------------------------------------------- |
> | Cosine Similarities | (Q1, Q2) = 95.10, (Q1, Q3) = 94.98, (Q2, Q3) = 89.90 |
> |  |  |  |
>
> **(3) Toward quality control in future work**: These metrics can also be adopted in future research as automatic filters to select or prune low-quality variants, further enhancing the robustness of CIT when deployed in practice.
>
> ------
>
> ### Q4: Why handle the two terms in Eq (3) separately, given that both involve the same evidence e?
>
> Thank you for this insightful question. We understand the concern and would like to clarify our interpretation and implementation.
>
> **（1）A clearer operational interpretation：** While both terms in Equation (3) involve the symbol e, they correspond to **distinct roles within the front-door decomposition**:
>
> - P(E=e∣Q=q) reflects the **observational process** of retrieving evidence from the question.
> - P(A=a∣E=e,Q=q′) estimates the model’s behavior under variant phrasing q′, **approximating a simulated intervention**.
>
> Although both involve e, their conditional contexts differ. We find that **treating them separately provides a clearer operational interpretation** and avoids mixing observational retrieval with counterfactual reasoning.
>
> **（2）adopt a tractable approximation:** Moreover, we recognize that theoretically, E is a random variable. Therefore, as described in Equation (5), we adopt a tractable approximation: we retrieve evidence from multiple semantically equivalent variants and **take their union as a surrogate for marginalization over E**.
>
> **（3）Empirical support:** This merged evidence set  is computationally tractable and has shown empirically strong performance (Table 5). While it is indeed an approximation of full marginalization, we think it remains **faithful to the spirit of front-door adjustment**, and provides a **stable and interpretable basis for causal estimation** in practical LLM-based TableQA scenarios.
>
> ------
>
> Meanwhile, thank you for pointing out the writing error—we will correct the notation issue in the revised version. We hope our responses have addressed your concerns. Thank you again for your thoughtful review. We would be glad to provide any further clarification or discussion if helpful.

---

> > ### Comment · Reviewer_fTKY · 2025-08-03
> >
> > Thank you for the response. I decide to keep my score.

---

### Official Review · Reviewer_SVNh · 2025-07-02

**Clarity:** 3
**Significance:** 2
**Originality:** 2
**Rating:** 4
**Confidence:** 3

**Summary:**

The paper proposes Causal Intervention TableQA (CIT) which uses a structural causal graph and front-door adjustment to reduce confounding caused by token co-occurrence biases. It achieves state-of-the-art performance across benchmarks, with consistent gains across various regimes confirming its generalizability.

**Questions:**

A key issue is the SCM could be misspecified. For the SCM, how to verify there is no edge between C and E? For example, the confounding may impact the table (i.e., evidence). How to verify there is no edge directly from Q to A? Any causal conclusion could be invalidated without rigorously verifying the specification.


For Figure 1 (left), why is the error caused by token co-occurrence? It seems more like a mismatch of complete information in the question instead (because it overly stresses on “exactly 12” and misses the keyword “Cambridge University”).


For experiments, why is the temperature set to 0.8? Would higher/lower temperature have an impact on the causal intervention? As a next step, how to mitigate hallucinated constraints?

**Ethical Concerns:**

["NO or VERY MINOR ethics concerns only"]

**Final Justification:**

My concern is addressed and I maintain my score.

**Limitations:**

Yes

**Quality:**

3

**Strengths And Weaknesses:**

The paper is well-written in general. The method of front-door adjustment looks novel and effective to address the key issue of token-occurrence in LLM-based table QA. The decomposition is simple in nature, yet each step of execution is grounded. The experiments are extensive on different baselines.


Weaknesses see questions.

---

> ### Author Rebuttal · Authors · 2025-07-30
>
> We thank the reviewer for the constructive feedback, encouraging recognition, and helpful questions. We are glad to clarify the raised points and respond in detail below.
>
> ---
>
> ### Q1: Potential misspecification of the SCM: how to justify no edge from C→E or direct Q→A link?
>
> **（1）On C→E**: We posit that the confounder C (token co-occurrence) influences evidence E only through its impact on Q. **In practice, evidence selection is conducted by prompting the LLM with Q;** hence, E is a downstream result of Q. There is no direct dependency between C and E in our pipeline, satisfying the front-door criterion.
>
> **（2）On Q→A**: The Q→E→A path is motivated by both empirical observation and established TableQA literature, and is further grounded in the Structural Causal Model (SCM), which underpins our front-door adjustment design. **Several prior works [1-5] formulate TableQA as below process: evidence retrieval conditioned on the question, followed by answer inference based on the retrieved content.** This supports our assumption that E fully mediates the influence of Q on A.
>
> - [1] Ye Y, Hui B, Yang M, et al. Large language models are versatile decomposers: Decomposing evidence and questions for table-based reasoning[C]//Proceedings of the 46th international ACM SIGIR conference on research and development in information retrieval. 2023: 174-184.
> - [2] Wang Z, Zhang H, Li C L, et al. Chain-of-Table: Evolving Tables in the Reasoning Chain for Table Understanding[C]//The Twelfth International Conference on Learning Representations.
> - [3] Lin W, Blloshmi R, Byrne B, et al. An inner table retriever for robust table question answering[C]//Proceedings of the 61st Annual Meeting of the Association for Computational Linguistics (Volume 1: Long Papers). 2023: 9909-9926.
> - [4] Wu J, Xu Y, Gao Y, et al. TACR: A Table-alignment-based Cell-selection and Reasoning Model for Hybrid Question-Answering[J]. arXiv preprint arXiv:2305.14682, 2023.
> - [5] Luo T, Lei F, Lei J, et al. Hrot: Hybrid prompt strategy and retrieval of thought for table-text hybrid question answering[J]. arXiv preprint arXiv:2309.12669, 2023.
>
> Although some recent methods adopt an end-to-end format, they often rely implicitly on latent retrieval steps within the model’s reasoning process. In contrast, our method explicitly enforces reasoning through E, ensuring that answers cannot be derived directly from Q alone. **This not only satisfies the front-door criterion, but also encourages step-by-step reasoning rather than shortcut predictions.**
>
> **（3）Empirical support**: We observe that when front-door intervention is disabled (e.g., by removing question variants in Table 5), model predictions degrade significantly, **indicating the presence of confounding and validating our causal formulation.**
>
> ---
>
> ### Q2: Figure 1 seems more like misunderstanding of “Cambridge University” than token co-occurrence.
>
> To ensure that the error shown in Figure 1 is not a coincidence tied to the presence of "Cambridge University," we provide multiple forms of empirical evidence to validate the presence of co-occurrence bias:
>
> - **（1）Adversarial perturbation (Figure 1)**: We apply double negation to logically equivalent questions in TabFact and observe a sharp performance drop. This demonstrates that **token-level perturbations, even without altering semantic meaning, significantly affect the model's prediction—indicating reliance on surface co-occurrence patterns rather than reasoning.**
> - **（2）Ablation on question variants (Table 5)**: When we remove question variant generation while keeping the question semantics intact, model performance declines. **This confirms that variants help mitigate co-occurrence bias**, which aligns with the function of front-door adjustment in our causal framework.
> - **（3）Support from existing literature**: Prior study [6] have shown that **increasing input diversity improves reasoning robustness and reduces overfitting to spurious cues**. These findings indirectly support our hypothesis and intervention strategy.
>   - [6] Maini P, Seto S, Bai R, et al. Rephrasing the Web: A Recipe for Compute and Data-Efficient Language Modeling[C]//Proceedings of the 62nd Annual Meeting of the Association for Computational Linguistics (Volume 1: Long Papers). 2024: 14044-14072.
>
> Together, these results confirm that the error in Figure 1 is not an isolated artifact, but rather a representative case of co-occurrence-induced confounding—one that our method is specifically designed to address.
>
> ---
>
> ### Q3: Why temperature=0.8? How does it affect intervention? How to mitigate hallucinated constraints?
>
> （1）**Follow prior studies** [7-8]: We chose temperature 0.8 based on prior studies, **as it provides a good trade-off between diversity and factual consistency**—both of which are essential for robust front-door estimation.
>
> - [7] Liu T, Wang F, Chen M. Rethinking Tabular Data Understanding with Large Language Models[C]//Proceedings of the 2024 Conference of the North American Chapter of the Association for Computational Linguistics: Human Language Technologies (Volume 1: Long Papers). 2024: 450-482.
> - [8] Yang Z, Du Z, Zhang M, et al. Triples as the Key: Structuring Makes Decomposition and Verification Easier in LLM-based TableQA[C]//The Thirteenth International Conference on Learning Representations. 2025.
>
> （2）**More results with other temperatures**: We further tested temperatures 0.4 and 1.0. Lower values lead to overly deterministic answers (hurting variant diversity), while higher values increase hallucinations. **Our empirical results show that 0.8 achieves the best balance.**
>
> | temperatures | result on WTQ |
> | ------------ | ------------- |
> | 0.4          | 75.62         |
> | 0.8          | **76.38**     |
> | 1.0          | 75.95         |
>
>
>
> （3）Hallucination is a well-known and broadly acknowledged challenge in current LLMs.
>
> - Our work does not aim to fully eliminate hallucinations, but rather to **constrain them in a principled way via causal intervention**. Specifically, We incorporate **Agent-based symbolic execution** and **majority voting** across multiple reasoning paths, which serve to regularize the model’s outputs and reduce overconfident or spurious answers.
>
> - As a next step, our method provides a **modular causal framework that is compatible with other hallucination mitigation strategies**. For example, it can be integrated with retrieval-augmented generation, symbolic consistency checks, or post-hoc verification modules to further enhance LLM robustness in TableQA.
>
> We sincerely thank you again for your valuable comments and insightful questions, which have helped us improve the clarity and rigor of our work. We hope our responses address your concerns, and we welcome further discussion if there are any remaining questions.

---

> > ### Comment · Reviewer_SVNh · 2025-08-08
> >
> > Thank you for the clarifications. Please incorporate them to your manuscript (if not). My concern is addressed and I have no follow-up questions. I will maintain my score.

---

### Official Review · Reviewer_MAtw · 2025-07-04

**Clarity:** 3
**Significance:** 4
**Originality:** 3
**Rating:** 5
**Confidence:** 3

**Summary:**

The paper identifies a key failure mode in TableQA with LLMs: reliance on spurious correlations arising from token co-occurrence in pretraining data. It introduces Causal Intervention TableQA (CIT), a framework that uses front-door adjustment to counteract latent confounding. CIT operates via: Question variant generation, Evidence aggregation, Dual-mode reasoning (Direct Prompting and Agent-based symbolic code execution), and Joint voting across outputs. Experiments on WikiTableQuestions, TabFact, and FetaQA show state-of-the-art results. The framework is tested across a variety of LLMs (open and closed source), showing robustness and generalization.

**Questions:**

Please refer to the Weaknesses part.

**Ethical Concerns:**

["NO or VERY MINOR ethics concerns only"]

**Final Justification:**

The author responses have addressed parts of my concerns. Therefore, I raised the score a bit.

**Limitations:**

yes

**Quality:**

3

**Strengths And Weaknesses:**

### Strengths

- The paper convincingly argues that LLMs often give unfaithful answers due to lexical biases and pretraining artifacts. The failure cases are real and well-demonstrated.

- Applying front-door adjustment in the TableQA context is novel and theoretically justified. The causal graph modeling (Q ← C → A) is intuitive.

- CIT outperforms competitive baselines across multiple datasets and models. The performance boost is nontrivial—especially the 6.2% absolute gain with GPT-4 and nearly 2% improvement over top baselines like TIDE.

- The paper includes ablations, breakdowns by question type, table size, and model scale, and evaluates robustness to adversarial perturbations. These analyses add significant value.

- The single-pass strategies for variant generation and reasoning reduce token/API cost while maintaining strong performance.

- Demonstrating gains on both closed and open-source models enhances credibility and practical relevance.

### Weaknesses

- The front-door adjustment is implemented in a heuristic way (i.e., generate variants, do majority vote), not through formal estimation of interventional distributions.

- The expression in Equation 3 is not estimated via probabilistic modeling; rather, it's approximated through aggregation/voting. Thus, the causal framing may feel more metaphorical than operational.

- Most components (variant generation, majority voting, agent reasoning) are reused from existing paradigms. The innovation lies in framing and orchestration rather than in novel modeling or algorithms. (I'm not saying that it cannot be innovative by putting together existing techniques. )

- Mix-SC and TIDE already explore DP + Agent combinations. The gains of CIT could partly stem from better prompt engineering or more voting, rather than causal reasoning.

- There is an inherent tension between latent confounders (which are unobserved) and front-door adjustability (which assumes a valid mediator). The assumptions behind the SCM and validity of the front-door criterion are not fully scrutinized.

- There is no proof that CIT truly eliminates the confounding bias—it’s mostly argued by performance gains.

- Accuracy or BLEU is used as a proxy, which doesn't directly test causal reasoning improvement.

- Some notational inconsistencies and redundant derivations across the main text and appendix could be tightened.

---

> ### Author Rebuttal · Authors · 2025-07-30
>
> We thank the reviewer for the detailed critiques. Below, we respond to each concern in turn.
>
> ### Q1: The front-door adjustment is implemented heuristically, without formal distributions.
>
> **(1) The black-box nature of closed-source LLMs makes probability estimation infeasible:** We agree that CIT approximates front-door adjustment through deterministic procedures rather than formal probabilistic modeling. This is a practical necessity, as the internal distributions of closed-source LLMs (only get token) are not observable or trainable. This limitation causes the use of structure-guided heuristics in place of full distributional estimation.
>
> **(2) CIT structurally follows the front-door criterion:** Although we do not explicitly compute interventional probabilities, our framework adheres to the structural decomposition of front-door adjustment:
>
> - Question variant generation approximates P(Q=q′);
> - Evidence aggregation estimates P(E∣Q=q);
> - Answer inference approximates P(A∣E=e,Q=q').
>
> This corresponds directly to Eq. 3. Each component in CIT plays the role of a counterpart in the front-door adjustment, ensuring that the **causal structure is faithfully preserved**, even if the operations are deterministic.
>
> **(3) Prior work has used causal as an effective heuristic:** Studies [1-2] have adopted causal not for exact identifiability, but as **heuristic guides** for system design in settings where exact confounder distributions or interventions are unavailable. Our work follows this tradition: we use the causal graph and front-door to **guide module design**, enable interpretability, and mitigate lexical bias via structured reasoning. The resulting gains support the practical value of this heuristic.
>
> - [1] Yang Z, Liu Y, Ouyang C. Causal Intervention-based Few-Shot Named Entity Recognition[C]//The 2023 Conference on Empirical Methods in Natural Language Processing.
> - [2] Ren L, Liu Y, Ouyang C. Causal inference-based debiasing framework for knowledge graph completion[C]//International semantic web conference. Cham: Springer Nature Switzerland, 2023: 328-347.
> ---
>
> ### Q2: The method primarily reuses known components.
>
> **(1) Causal framing provides theoretical foundations that facilitate future method development:** Our work is the first to introduce causal intervention into LLM-based TableQA, framing token co-occurrence as a confounder and modeling the task using a structural causal graph. This formulation not only justifies our design of question variants and evidence mediation, but also **provides a principled foundation for future methods to build upon and make theoretically grounded improvements.**
>
> **(2) Existing components are leveraged to ensure effective and efficient implementation:** We use mature components to ensure that CIT inherits the best practices of existing paradigms, while being driven by a novel causal objective: mitigating bias induced by token-level confounding. As our results shown, this combination leads to **consistent performance gains**, validating the effectiveness of causal guidance even when applied through standard building blocks.
>
> ---
>
> ### Q3: Improvements may stem from better prompting or voting.
>
> **(1) Prompts are consistent with prior work:** We largely adopt the same prompting strategies as Mix-SC and TIDE for both DP, PyAgent and voting. Despite this, CIT achieves SOTA, suggesting that the gains stem from our **causal design rather than prompt tuning**.
>
> **(2) Adversarial results(Figure 1) show that CIT mitigates confounding bias:** When using identical components and prompts, models without CIT exhibit significant results degradation under double negation perturbations. In contrast, **CIT remains robust**, indicating that its effectiveness arises from explicitly addressing confounding via causal intervention.
>
> **(3) Removing causal components leads to performance drops:** Ablation results (Table 5) demonstrate that removing key components related to front-door —such as variant generation— leads to notable accuracy degradation. This confirms the **functional role of causal components in enhancing robustness.**
>
> Overall, the strong performance of CIT cannot be explained by prompt alone; rather, it results from **causally grounded mitigation of confounding bias** inherent in task.
>
> ---
>
> ### Q4: Tension between latent confounders and front-door assumptions; is the SCM valid?
>
> **(1) Our SCM is carefully constructed based on the TableQA task pipeline and LLM behavior.**
> This causal graph aligns with how LLMs use evidence in the reasoning process [3–7], and it satisfies the structural requirements of SCMs. While we acknowledge that SCM is an abstraction, it is grounded in actual system behavior and has solid theoretical foundations.
>
> - [3] Ye Y, Hui B, Yang M, et al. Large language models are versatile decomposers: Decomposing evidence and questions for table-based reasoning[C]//Proceedings of the 46th international ACM SIGIR conference on research and development in information retrieval. 2023: 174-184.
> - [4] Wang Z, Zhang H, Li C L, et al. Chain-of-Table: Evolving Tables in the Reasoning Chain for Table Understanding[C]//The Twelfth International Conference on Learning Representations.
> - [5] Lin W, Blloshmi R, Byrne B, et al. An inner table retriever for robust table question answering[C]//Proceedings of the 61st Annual Meeting of the Association for Computational Linguistics (Volume 1: Long Papers). 2023: 9909-9926.
> - [6] Wu J, Xu Y, Gao Y, et al. TACR: A Table-alignment-based Cell-selection and Reasoning Model for Hybrid Question-Answering[J]. arXiv preprint arXiv:2305.14682, 2023.
> - [7] Luo T, Lei F, Lei J, et al. Hrot: Hybrid prompt strategy and retrieval of thought for table-text hybrid question answering[J]. arXiv preprint arXiv:2309.12669, 2023.
>
> **(2) The front-door adjustment is applicable in our setting and is empirically supported.**
> In the SCM framework, back-door adjustment requires the confounder C to be observable. In contrast, **front-door adjustment allows C to be unobserved by introducing a mediator E and intervening through the do-operator, which blocks the confounding path Q←C→A**. This ensures the validity of causal inference. The formal justification can be found in [8–9] and our Appendix B.
>
> - [8] Pearl J, Glymour M, Jewell N P. Causal inference in statistics: A primer[M]. John Wiley & Sons, 2016.
> - [9] Norman E. Fenton, Martin Neil, and Anthony C. Constantinou. The book of why: The new science of cause and effect, judea pearl, dana mackenzie. basic books (2018). *Artif. Intell.*, 284:103286, 2020
>
> **(3) Front-door adjustment has been shown effective in many studies for handling unobserved confounders and improving performance.**
> Inspired by these findings, our method incorporates a mediating path to avoid dependence on unobservable variables and enables more robust causal estimation.
>
> - [10] Yang X, Zhang H, Qi G, et al. Causal attention for vision-language tasks[C]//Proceedings of the IEEE/CVF conference on computer vision and pattern recognition. 2021: 9847-9857.
> - [11] Yang Z, Liu Y, Ouyang C, et al. Improving few-shot named entity recognition with causal interventions[J]. Big Data Mining and Analytics, 2024.
>
> **(4) We also provide empirical evidence supporting the validity of our causal structure:**
> Removing the evidence module or question variants leads to a significant drop in performance (Table 5);
> In addition, adversarial examples (Figure 1) show that CIT effectively mitigates the confounding bias caused by token co-occurrence, whereas other methods fail under the same setting.
>
> In summary, both the SCM construction and the application of front-door adjustment in our method are theoretically sound and empirically validated.
>
> ---
>
> ### Q5: No direct evidence that CIT eliminates confounding bias.
>
> We acknowledge the challenge of measuring latent confounding. Nevertheless, we offer **three complementary forms of indirect evidence**:
>
> - **Adversarial robustness**: Double negation examples cause large drops for vanilla models, but CIT is more stable;
> - **Ablations**: Removing causal front-door components degrades results;
> - **Model transferability**: CIT improves across open-/closed-source LLMs (Table 6), suggesting bias reduction generalizes.
>
> These collectively support the bias mitigation effect.
>
> ---
>
> ### Q6: Evaluation metrics are indirect; causal reasoning is not directly tested.
>
> (1) First, we adopt **Accuracy and BLEU as our primary metrics to ensure consistency and fair comparison with prior LLM-based TableQA work [12-13].**
>
> - [12] Liu T, Wang F, Chen M. Rethinking Tabular Data Understanding with Large Language Models[C]//Proceedings of the 2024 Conference of the North American Chapter of the Association for Computational Linguistics: Human Language Technologies (Volume 1: Long Papers). 2024: 450-482.
> - [13] Yang Z, Du Z, Zhang M, et al. Triples as the Key: Structuring Makes Decomposition and Verification Easier in LLM-based TableQA[C]//The Thirteenth International Conference on Learning Representations. 2025.
>
> (2) While we acknowledge the **absence of ground-truth causal labels and the difficulty of directly measuring latent confounding**, we provide **three complementary forms of indirect evidence** supporting the bias mitigation effect:
>
> - **Adversarial robustness**: Double negation examples lead to substantial performance drops for baseline models, whereas CIT remains stable (Figure 1);
> - **Ablation studies**: Removing components related to front-door adjustment—such as question variants—significantly degrades performance (Table 5);
> - **Model transferability**: CIT yields consistent gains across both open- and closed-source LLMs (Table 6), suggesting that the reduced confounding bias generalizes across model families.
>
> Together, these results offer strong indirect support for the causal benefits of CIT.
>
> ---
>
> We hope these responses address your concerns. We are grateful for your feedback and would welcome further discussion if needed.

---

> > ### Comment · Reviewer_MAtw · 2025-08-04
> >
> > The author responses have addressed parts of my concerns. I have raised the score a bit.

---

### Decision · Program_Chairs · 2025-09-17

**Decision:**

Accept (poster)

**Comment:**

This paper introduces Causal Intervention TableQA (CIT), a framework that addresses a fundamental failure mode in LLM-based TableQA: reliance on spurious correlations arising from token co-occurrence in pretraining data. By framing TableQA within a structural causal model and applying a front-door adjustment strategy through question variant generation, evidence aggregation, dual-mode reasoning, and joint voting, the paper provides both a principled causal perspective and a practical solution.

The strengths of this work are clear. Reviewers unanimously highlight that the problem is well-motivated and convincingly demonstrated. The causal framing is novel in the TableQA context, and while many of the components draw on existing paradigms, the orchestration guided by causal principles leads to consistent and significant improvements. CIT achieves state-of-the-art performance across multiple benchmarks (WikiTableQuestions, TabFact, FetaQA) and models (both open- and closed-source), with strong ablations, adversarial robustness experiments, and analyses across question types and table sizes. Importantly, the author rebuttal clarified concerns about the causal assumptions, the heuristic implementation of front-door adjustment, and the role of question variants, and provided additional experimental evidence that CIT’s gains cannot be explained by prompt tuning or voting alone.

The weaknesses raised—such as the heuristic nature of front-door adjustment, reliance on variant generation quality, and remaining challenges in numerical reasoning—are valid but do not detract from the paper’s overall contribution. The authors’ rebuttal addressed these points thoroughly, situating their work within the tradition of using causal framing as a heuristic guide when full probability estimation is infeasible. Reviewers agreed that the responses strengthened the case for acceptance, with most maintaining or raising their scores.